# Experiences of cervical cancer survivors in Chitwan, Nepal: A qualitative study

**Gambhir Shrestha**[1]*, **Rashmi Mulmi**[2], **Prabin Phuyal**[3], **Rahul Kumar Thakur**[3], **Bhola Siwakoti**[2]

1 Department of Community Medicine, Maharajgunj Medical Campus, Institute of Medicine, Tribhuvan University, Maharajgunj, Kathmandu, Nepal, 2 Department of Cancer Prevention, Control and Research, B. P. Koirala Memorial Cancer Hospital, Bharatpur, Chitwan, Nepal, 3 B.P. Koirala Institute of Health Sciences, Dharan, Sunsari, Nepal

* gamvir.stha@gmail.com

## Abstract

### Introduction

Cervical cancer is a global leading cause of morbidity and mortality. The majority of cervical cancer deaths occur in developing countries including Nepal. Though knowledge of cervical cancer is an important determinant of women's participation in prevention and screening for cervical cancer, little is known about this topic in Nepal. This study explores the experiences of cervical cancer survivors and assesses the attitude of family and community towards it and stigma related to this disease in Bharatpur, Nepal.

### Methods

The study design was qualitative methods involving two focus-group discussions. A total of 17 cervical cancer survivors, who have completed two years of cancer treatment were selected purposively from Chitwan. All qualitative data were transcribed and translated into English and were thematically analyzed.

### Results

The majority of the participants had scant knowledge about cervical cancer, its causative agent, showed less cervical cancer screening, delayed healthcare-seeking behavior despite having persistent symptoms before the diagnosis. The main reasons identified for not uptaking the cervical screening methods were an embarrassment and having no symptoms at all. Most of them endured social stigma related to cervical cancer in the form of physical isolation and verbal abuse.

### Conclusions

There is an urgent need for interventions to make women and the public aware of cervical cancer and launch effective health education campaigns, policies for cervical cancer prevention programs. This implementation can save the lives of hundreds of women and help them avoid going through all the negative experiences related to cervical cancer. More

**Data Availability Statement:** All relevant data are within the paper.

**Funding:** The authors received no specific funding for this work.

**Competing interests:** The authors have declared that no competing interests exist.

studies are required to gain the perspectives, knowledge, experiences, and attitudes of cervical cancer survivors to add to the research.

## Introduction

Cervical cancer is the second most common cancer in females mainly in low and middle-income countries. Around 310,000 deaths occur annually due to cervical cancer [1]. Human papillomavirus (HPV) infection is the major cause of cervical cancer. Although most cases of HPV will resolve on their own, persistent infection with certain types of HPV (types 16 and 18) on the cervix can lead to precancerous lesions that can progress to cervical cancer [2]. Nepal has a population of 10.1 million women aged 15 years and older who are at higher risk of acquiring cervical cancer. Cervical cancer is the most frequent cancer among women between 15–44 years and it also ranks as the first most frequent cancer among women in Nepal. According to the latest data of 2018 around 1,928 women die yearly out of a total of 2,942 women who are diagnosed with cervical cancer yearly in Nepal [3].

Early detection of HPV infection through the utilization of Papanicolaou (Pap) testing has been shown to decrease rates of cervical cancer [4]. Because of restricted access to the health care facility and limited knowledge about the preventive techniques for cervical malignancy, the greater part of the women in Nepal has never had a pap test in entire life. Women have reported a lack of knowledge of cancer screening, limited health care facilities, lack of time and money, logistics barriers, and lack of social support as obstacles to receiving regular cancer screening services [5, 6]. Since early-stage cervical cancer is curable, it is significantly important to focus on the methods to improvise the quality of life of such patients [7, 8]. The five-year survival rate for cervical cancer is around 68%, in developed countries, but is very less in developing countries [9].

Females diagnosed with cervical cancer often face many difficulties associated with cancer itself, cancer treatments and their side-effects, social understandings, and monetary constraints [10]. With a very strong association of cervical cancer with sexual practices in women, cervical cancer has a very strong impact on the quality of life of cancer survivors [11]. Social shame and the absence of satisfactory information concerning the improvement of cervical disease have supported a low quality of life. Besides, cervical cancer patients have been found to have more terrible personal satisfaction scores when contrasted with everyone as well as when contrasted with other gynecological malignancy survivors [12].

A very minimal subjective investigation has been done to draw out the experience of the malignancy survivor in Nepal. The objective of the study is to understand how a survivor of cervical cancer and their caregivers understood, experienced, and were impacted by stigma as well as their perspectives on how to measure and intervene to reduce this stigma.

## Methods

### Study design and data collection

The qualitative research design using focus group discussion (FGD) was used to explore the experiences of cervical cancer survivors. Two FGDs were conducted, consisting of 8 members in one and 9 members in the other. The study was conducted from January 2019 to October 2019. An interview guide was prepared by an expert panel of faculty with the help of already published articles [10, 13]. FGD was conducted in the Nepali language using the interview guide by the principal investigator (GS) along with a female Nepalese researcher (RM). The principal investigator conducting the FGD had a postgraduate degree in community medicine

with experience in qualitative research. The FGDs were conducted in the meeting room within B.P. Koirala Memorial Cancer Hospital (BPKMCH) premises. An initial introduction was done among the participants and the research staff before beginning the FGD session. The FGD was facilitated by GS and RM. The questionnaire consisted of open questions regarding personal experiences with cervical cancer, perceptions of what might have caused their cervical cancer, experience with any symptoms, Pap screening, diagnosis, and treatment, and if there were any barriers experienced in cervical cancer screening. FGD was conducted over 60–90 minutes. With the help of two moderators, the FGDs were conducted. All the participants were comfortable with the Nepali language, hence the Nepali language was used for the discussion. At first, information regarding sociodemographic factors was collected. All the discussion was audio tape recorded for further translation and transcriptions. Similarly, all the participants were encouraged to share their knowledge, belief, and stigmata in society as experienced by them. Further, a dedicated note-taker took detailed field notes to complement the audio recorded FGDs. To maintain confidentiality and privacy of the participant's information, it was ensured none other than the participants and the moderators attended the sessions.

## Participant selection

The inclusion criteria were any cervical cancer survivors residing nearby BPKMCH, who have completed more than two years of cancer treatment. They were selected purposively and contacted via telephone to participate in the study. A total of 17 participants who agreed to participate in the research were invited to BPKMCH for further study. Only two FGDs were possible as obtaining the cancer survivors within the vicinity of the Cancer Hospital was limited.

## Data analysis

After the collection of data, the audio records and notes were translated and transcribed from the Nepali language to the English language by two members of the research team (GS, RM). Transcripts were examined line-by-line and analyzed using the framework approach. The themes were taken based on previous studies. This approach was chosen, as it allows deductive analysis based on our study objectives. Three members of the research team (GS, RM and PP) independently coded the FGD through discussion and agreed on the coding framework. The data was organized into charts. This helped to put the analyses of responses according to the participants. The socio-demographic characteristics of the survivors were presented in frequency, percentage, mean and standard deviation.

## Ethics approval and consent to participate

The study protocol was approved by the Ethical Review Board of Nepal Health Research Council (Reg. No. 152/2019). All participants were explained about the research objectives, their expected role, and the voluntary nature of participation. They were also informed that their decision to participate or decline participation would not affect any benefits or services received by them. A written informed consent (if literate) or thumbprint in presence of a witness (if illiterate) was obtained before participation in the study and permission to audio record the conversation was taken.

## Results

### Socio-demographic characteristics of the survivors

The mean age of the participants was 53 years (SD 10.8) ranging from 35 to 81 years. On average, the age of diagnosis of cervical cancer among the participants was 47 years (SD 8.7). In

**Table 1. Socio-demographic characteristics of participants.**

| Variables | Frequency (n) | Percentage (%) |
|---|---|---|
| **Age group (in years)** | | |
| 35–50 | 7 | 41.2 |
| 51–65 | 9 | 52.9 |
| 66+ | 1 | 5.9 |
| **Marital status** | | |
| Married | 15 | 88.2 |
| Widowed | 2 | 11.8 |
| **Religion** | | |
| Hindu | 12 | 70.6 |
| Buddhist | 4 | 23.5 |
| Christian | 1 | 5.9 |
| **Level of education** | | |
| Illiterate | 11 | 64.7 |
| Primary | 2 | 11.8 |
| Secondary | 4 | 23.5 |
| **Family history of Cancer** | | |
| Yes | 4 | 23.5 |
| No | 13 | 76.5 |

our study, all the participants were ever married with 88% currently married and 12% widowed. The majority (70.6%) were Hindu by religion followed by Buddhism (23.5%) and Christianity (5.9%). Most (64.7%) of the participants were illiterate. A quarter of the participants had a history of cancer in their families (Table 1).

## Awareness of survivors about cervical cancer

In each group, survivors were asked about their idea of how they might have acquired cervical cancer. All the survivors shared their knowledge about the cause of cervical cancer. The majority of the women responded that they truly had no idea regarding the cause of cervical cancer (Table 2). Women described cancer as a painful experience they ever had. As one survivor stated,

**Table 2. Summary of the survivor's reflection of the cause of cervical cancer.**

| Theme | Representative quote |
|---|---|
| Women's fair skin color | *"Well, I guess it is more common in fair women or you know like it may be due to short birth spacing. But exactly I don't know the cause of it [cervical cancer]."* [46-year-old] |
| Short birth spacing | |
| Use of tobacco products | *"I heard somewhere that smoking and chewing tobacco causes cancer. Well, I don't have those habits still I got cervical cancer. I know that smoking and chewing tobacco causes cancer but, I wonder what caused cancer in me. . . ."* [45-year-old] |
| Consumption of unhealthy diet | *"I think that it [cervical cancer] is related to food. I mean to say that there must be something bad in the food we eat that predisposes us to cancer. Or you know, it may be due to bearing too many children. And also, women are physically weak which might be one of the reasons for us getting cancer."* [62-year-old] |
| Bearing too many children | |
| Physical weak gender | |
| Use of homemade menstrual cloths | *"Growing up in a village, I did not have much knowledge about health and sanitation. During my periods, I used homemade menstrual cloths. I guess I got cancer from those dirty pieces of cloths which I used during my periods. From my experience, I believe careless in menstrual hygiene causes cervical cancer."* [59-year-old] |

*". . . . Look, I have no idea what the cause behind my cervical cancer is. It came out of nowhere. It was such a devastating time when I discovered I had cervical cancer. I thought I was dying. All I did was sit and ask myself why. . . . why it happened to me, did I do anything wrong?". . . .* [60-year-old]

While the majority remained clueless, few of them came along with some ideas of how they might have acquired cervical cancer. The ideas about getting cervical cancer varied among individuals and included fair skin color, short birth spacing, bearing too many children, unhealthy dietary habits, smoking and chewing tobacco, and the use of homemade menstrual cloths. One of the survivors even referred to the physical weakness in female as one of the causes of cervical cancer. When they were further asked specifically if they knew about the human papillomavirus (HPV), most of the survivors responded that they had never heard about the virus. Most of them did not know about the virus as the main causative agent of cervical cancer, while two of them came to know about the virus from their doctors after they discovered their cervical cancer.

### Symptoms they experienced prior to diagnosis

**Episodes of vaginal bleeding.**    All the survivors were enquired about the symptoms they experienced before they were diagnosed with cervical cancer. The majority of the women reported that they had episodes of unexplained vaginal bleeding. One of them even attributed the bleeding episode to hemorrhoids.

*"One day when I was in the bathroom, I saw blood in the pan, but falsely I attributed it to be from the hemorrhoids. I was not aware of what that could mean, but to be safe I went to see a doctor. After examination, the doctor told me that something isn't right and recommended me to undergo a cervical biopsy. After a couple of days, I found out I had cervical cancer. He [the doctor] suggested me for the removal of my uterus, but I was reluctant to undergo surgery. Then I underwent an imaging test [MRI], and thanks to God my cancer had not spread to my surrounding organs [urinary bladder], and given the options, I opted for the radiotherapy"* [46-year-old]

**Attributing vaginal bleeding to physiological changes.**    Many times, women attributed their episodes of vaginal bleeding to the post-menopausal changes. Thinking bleeding as a physiological change that can occur at any time after they have reached menopause, many women did not seek immediate medical consultation.

*"At age 45, I had an episode of vaginal bleed and I discussed it with my friends. They believed that this [vaginal bleed] is pretty much common after you attain menopause and there is no need for any medical consultation. But later, I had an episode of huge bleeding [from the vagina] and I was totally scared. Fearing there might be something wrong going on, I went for the medical consultation right away. The doctor described me as I had some kind of wound in my cervix and urged me to undergo a cervical biopsy. Later, the report came as cervical cancer and I got operated and got my uterus removed."* [54-year-old]

*"As far as I can remember, I did not bleed for long after I reached menopause. And suddenly out of nowhere, I started having vaginal bleeding. I used to see blood even while passing urine. It was hot weather, which I believed was the reason behind my bleeding. It did not improve for long which made me worried. I went to meet the doctor. After all the examinations and tests like a biopsy, it came out to be cervical cancer."* [81-year-old]

**Acute severe vaginal bleeding.** In addition to the above symptoms in which women did not seek immediate medical consultation, one of the women had heavy vaginal bleeding which brought her to the emergent medical attention, and through this, her cervical cancer was diagnosed.

*"I used to have lots of episodes of vaginal bleeding, which did not bother me initially. Later, I had a heavy bleeding episode and even had clots in the bleed. It was like blood everywhere. Then without thinking much, I decided to see a doctor. I was referred to a cancer specialist from a local hospital. They [doctors] told me to get a cervical biopsy right away and asked me to see them in a week. My son collected my reports later and the opposite of what I expected the results came back as positive for cancer. That moment is indescribable."* [60-year-old]

**On and off abdominal pain.** One of the women reported that she had on and off abdominal pain which she could not attribute to cervical cancer. She vividly recalled the event when she was diagnosed with cancer as devastating news for her.

*"Initially, I used to have pain, on and off, in my lower belly and I took medicine from the nearby pharmacy thinking it as a urinary tract infection. I don't remember for how long I had that [abdominal pain]. Later, I was advised to see a doctor. During an appointment, he [the doctor] told me my reports were concerning, and after biopsy, it turned out to be cervical cancer. The diagnosis hit me very hard. I felt my world spinning around me when I knew I had cancer. I cried a lot. I had no idea whether it is a genetic or something else."* [49-year-old]

## Barriers to cervical cancer screening

Women were enquired about their experiences with the Pap screening before getting diagnosed with cervical cancer. Surprisingly, none of them ever had any type of cervical cancer screening before they were diagnosed with cervical cancer. All of them avoided screening in the past because of embarrassment or shame. As reported by them, women feel shy to expose their private parts to doctors.

*"Well, I have to say most of the women feel too shy to go for screening and expose their private parts to the medical professionals. Regarding me, I am not much comfortable with it, like, how can I show my private parts to the people other than my husband."* [61-year-old]

*"To be honest (smiling), women are not comfortable showing their private parts to others. Even when we know about the medical camps in the nearby hospital or health post, we pretend to be fine if we were asked by someone else and avoid going for the screening programs. . . ."* [45-year-old]

Feeling healthy or having no symptoms at all was also mentioned to be one reason for women to not go for screening.

*"I had a very bad experience. Back in time about 10 years ago, a cervical screening camp was organized in our village. Without thinking much about it, I did not go there for the screening, as I was feeling well and believed nothing could be wrong going with me. But what I did was a mistake. Later I got cancer and had to undergo surgery. If I had been careful enough and had undergone the routine screening process, I could have avoided surgery and all the difficulties associated with it. I am also a political leader in my ward so, I conducted a survey and you will be surprised knowing that 72 out of 100 females did not have their routine screening done."* [59-year-old]

## Cervical cancer in Nepalese society

Women were enquired about how their family members and society treated them after knowing that she had cervical cancer. The responses we got varied among individuals. Some of them were well treated by their family and society, while others endured the stigma associated with cervical cancer. The majority of the participants believed that even in present-day cervical cancer is stigmatized in the Nepalese society because of the misconceptions people have towards cancer as a whole (Table 3).

**Drivers of stigma.**    According to the women, the general themes acting as the driving factors for cancer stigma in society are the people's perception of cancer as an ultimate death result and cancer as a communicable disease. Cancer as a whole is stereotyped as a cause of death and is contagious in society and despite providing clear information by the affected person, most of the time it was hard to change their views. It is due to the lack of clear information regarding cancer, such beliefs do exist in society. As the respondents mentioned,

> *"When the people in my neighborhood knew I had cervical cancer, they all [neighbors] believed that it [cervical cancer] is a communicable disease and thought like you know, cancer is incurable, I am not going to live for long."* [60-year-old]

> *"The very first thing I used to hear from them [relatives] was "You are going to die soon from cancer". In return, I used to give them the information about my disease and say that this is curable and I would be fine. But they did not believe me."* [47-year-old]

**Manifestation of stigma.**    The fear that existed among the society members that it can be transmitted via casual close contact and communication lead to the physical isolation or marginalization of the cancer patients in their family as well as in the society. According to respondents, neighbors started avoided coming close to them or inviting them in social events, and even verbally abused and pitied them. Most of the time, they felt that they were hated by society just because they had cancer. They mentioned how physical isolation and marginalization resulted in the loss of their social support.

> *"When I was undergoing chemotherapy for my cancer, my neighbors stopped coming near to me thinking that they would get cancer from me. Even my family members wore a mask when they had to come near me."* [60-year-old]

> *"Back in our time, people from our village used to take cancer in a very negative way. When I got cancer, they [villagers] didn't even allow me to go near them and used to say "go away" if I attempted to do so."* [81-year-old]

> *"I had a very hard time dealing with my diagnosis of cervical cancer. That was the first time anyone outside my family came to see me and pitied me. During my chemotherapy, I lost my hair and my neighbors believed it was happening due to my cancer. I don't know why but they [neighbors] always gave me a disgusting look when I said I am undergoing radiotherapy. I felt like everyone hated me."* [47-year-old]

**Table 3. Summary of the stigma related to cervical cancer.**

| Drivers of stigma | Manifestation of stigma | Consequences of stigma |
|---|---|---|
| Cancer as a death | Physical or social isolation | Loss of social support |
| Cancer is contagious | Verbal abuse | |
| | Self-stigma of fear | |

**Self-stigma of fear associated with cancer.**   The fear related to cancer and its treatment also existed in those living with cancer. Due to the negative connotations of cancer, women having cancer often feared to undergo the treatment. As one of the respondents mentioned,

*"Initially I was reluctant to undergo the treatment of my cancer [cervical cancer], as one my friend told me not to go for treatment as cancer will eventually kill me and also the radiotherapy is very difficult to undergo. Later I met a woman with cervical cancer who encouraged me to undergo treatment. After hearing her experience my perspective towards chemotherapy and radiotherapy changed. Meeting her eased my fears. I am grateful that I met her, otherwise, I would not have gone for the treatment and only God would know what would have happened to me. . .hahaha. . . .."* [40-year-old]

**Social support and family encouragement.**   According to them, the love, care, and support they got from their family and friends were the key elements for them to fight cancer and stay optimistic through the journey of cancer.

*"I had very good support from my friends and family (smiling). They treated me very well when I got cancer and was undergoing treatment. I am so glad that they were all so supportive and with me when I needed them the most."* [46-year-old]

*"My family looked after me very nicely. I can't say enough about them who cared for me during my hard time. Now, I love them more than ever. They are the most important person in my life."* [45-year-old]

*"Yes, family support played a great role in my survival. I do not have enough words to express my gratitude for the care, help, and love I got from them. They were always by my side I needed them."* [59-year-old]

According to a few respondents, society's perception overturned with time as they saw the patients with cervical cancer doing well with the treatment and also with the increasing awareness via media. As one of the survivors stated,

*"At present, the situation is quite different than the previous one. With the awareness arising from the media and campaigns, their perspective towards cancer changed. Now they [society] feel ashamed of how they previously treated cancer patients. These day people are very supportive towards the cancer patients. . . . . ."* [81-year-old]

## Unexpected misconceptions

A few uncommon views regarding the cancer were identified from our interview. People have few misconceptions regarding the type of food they should take and the use of herbal medicine. They avoided a certain type of food during their period of cancer. One of the survivors avoided meat and eggs, while one patient consumed only pulses, milk, and fruits. Among them, one took some herbal medicines recommended by her friend, which she does not know about.

*"I did not like to eat for a month. I just eat pulses, soups, milk, and fruits only during my treatment. Sometimes I used to panic a lot and pray to god for my survival. I believe I would not survive if God was not there. . . . . ."* [60-year-old]

*"I did not eat meat for 1 year and also the eggs. . . . . ."* [61-year-old]

*"One of my neighbors was diagnosed with some serious disease, I don't know exactly what it was, but I know is she was doing fine by taking some herbal medicines. She named about 13 types of herbal medicines she was taking. Seeing her I also started taking those [herbal meds] on daily basis. I believe that also gave great benefits to my health."* [53-year-old]

### Message of the survivors

The majority of the women did not go for the Pap screening for cervical cancer, and also after being diagnosed few of them were reluctant to undergo treatment. However, though lately, but with their own experiences with cancer, they realized the importance of screening methods for early detection and treatment of cervical cancer.

*"I heard about one cancer patient who did not undergo radiation therapy because of all the negative comments she heard about radiotherapy, its complications. I don't know where she is now. That is certainly a mistake. Please don't do that, please go and get your treatment before it is a delay so that you don't have to regret it forever."* [51-year-old]

*"Today I am living a new life. For those women out there who are healthy, I suggest them to go for screening regularly. And to those living with cervical cancer, don't worry much about it. Yes, cancer is a disease but is not always a killer; they can be cured with treatment."* [60-year-old]

Thus, they advised other healthy women to undergo regular screening for early detection and undergo treatment if they are diagnosed case of cervical cancer.

## Discussion

This qualitative study done to explore the experiences of cervical cancer survivors is the first of its type that had been conducted in Nepal. Survivors expressed their perspectives of cervical cancer and its cause, symptoms they had prior to diagnosis, their cervical cancer screening behavior, and the personal and social consequences they had faced after the diagnosis. Several findings can be derived from the study.

### Paucity of information

The majority of women we interviewed, had scant information about the cause of cervical cancer, and almost all of them had never heard about HPV before. Among the participants, 64.7% of the women are illiterate, and illiteracy is perhaps the reason behind their lack of knowledge about cervical cancer. Several other studies in Nepal reported a lower level of knowledge in women about cervical cancer. For instance, in a study in Rukum, Nepal only 5% of the 600 participants had ever heard about cervical cancer and only 30% among themselves were able to relate HPV as its causative agent and concluded their poor awareness about cervical cancer to illiteracy and lower household income [14]. In another study in Eastern Nepal, only 21.5% of women could identify HPV as a causative agent of cervical cancer [15], and another study in rural Nepal reported 76.24% of women did not know anything about cervical cancer [16]. Similarly, the lower degree of awareness of cervical cancer was reported in a study among cervical cancer survivors in North Carolina, USA [10]. This highlights the importance of implementing educational programs and campaigns to educate women and make them aware of cervical cancer.

## Barriers to cervical cancer screening

The barriers to cervical cancer screening identified from the study are feeling well or having no symptoms at all and embarrassment that results from showing their private parts to the health professionals. In the same way, previous studies reported illiteracy [14, 17, 18], lack of awareness about cervical cancer and its screening methods, embarrassment, having no symptoms at all [19, 20], fear of finding out cancer [16], sociocultural barriers, service providers' behavior, geographical challenges, poor financial condition [20] as significant barriers to cervical cancer screening uptake behavior in Nepal. On contrary, living in a rural area [18], participating in awareness programs, support from family and women's group [20] were shown to be facilitators to increase the screening uptake in women. Similarly, a study in Morocco also reported poor awareness about cervical cancer and having no symptoms at all as a cause for poor Pap screening uptake practice [21]. Moreover, stigma related to cervical cancer was a major barrier to cervical cancer screening in Karnataka, India [13], and poor finances and lack of insurance, sense of well-being, transportation issues, and dissatisfaction with the behavior of service provider were related to poor screening behavior among survivors in North Carolina, USA [10]. Prior bad experiences with the detection of cancer were also described as a barrier to cervical cancer screening among older African American and Hispanic urban women [22].

The findings suggested the need for interventions at multiple levels to increase the screening uptake behavior in women. First, women should be made aware of the fact that early detection of cervical cancer from regular screening and timely treatment can significantly save women's lives. Second, there seems to be a need for the implementation of cervical cancer screening programs in Nepal to make it accessible to women in all the geographical areas and cheap enough to make it affordable. Awareness programs via media, mass campaigns, healthcare professionals all can equally increase the screening uptake behavior in women. As per the literature, educating women about the importance of screening [23, 24] and using the newer information technologies, like reminders and messages via the cellphones [25] or involving the social media influencers [26] for increasing awareness have shown increased participation of women in a cervical cancer screening program.

## Unconcerned towards the symptoms

The majority of women demonstrated delayed healthcare-seeking behavior despite experiencing symptoms for a long. As per our results, the women's sense of well-being and their inability to relate their symptoms to a possible serious disease or attributing their symptoms to normal physiological changes like menopause possibly explain the delayed healthcare-seeking behavior. Similarly, delayed healthcare-seeking behavior despite persistent symptoms was also noted in cervical cancer survivors in North Carolina [10]. On the other hand, some studies articulated the delayed healthcare seeking to the stigma related to cervical cancer [13]. This behavior seems concerning as this frequently leads to delayed presentation to medical attention thus, leading to delayed diagnosis and treatment which can increase the negative impact of disease in affected individuals. The women should be educated about the fact that the sense of well-being does not ensure everything going alright in them and should never be used to rule out the need for routine screening measures or medical consultation. Women should be educated about the symptoms related to cancer, encouraged to seek medical care when they had any symptoms. Likewise, effective communication seems important between the health-care providers, policymakers, and the public via educational programs, and media to improve the knowledge about cancer in women.

## Manifestation of stigma related to cervical cancer

Even in today's modern era, cervical cancer is still being stigmatized in our society. It is quite surprising that despite being such a highly prevalent disease, the public has misunderstood cervical cancer, and has different perspectives about it. The misconceptions regarding cervical cancer seem to originate stigma related to cervical cancer and keep perpetuating it. Our study revealed that pertaining cervical cancer to ultimate death and fear of transmission assuming it as a communicable disease are the driving factors for the origin of stigma. The perceptions that are driving the stigma related to cervical cancer such as cancer is a communicable disease and leads to ultimate death are described in studies conducted in India [13] and in Brazil [27]. Furthermore, the mind of the people is constantly preoccupied with ideas of attributing cervical cancer to the individual's promiscuous sexual behavior, like sleeping with multiple men, or relating it to sexually transmitted disease [27] or blaming cervical cancer as a punishment to individual's bad deeds/transgressions, like assuming they must have done something wrong [13], and these reasons naturally rooted the stigma related to cervical cancer. Blaming cervical cancer to the promiscuous sexual behavior was also described as a driving factor for stigma related to cervical cancer in studies conducted in America and Zambia. society [28–30].

The stigmas associated with cancer manifested in various forms in the communities and women had to endure the discrimination and negative impacts arising because of it. Although all the women were not the victim of stigma, the majority of the interviewees described the enacted stigma as discrimination in the form of social isolation or marginalization ranging from community people no longer coming in close contact to the person with cancer, inviting them to participate in social events using masks by the family members while coming in contact to them. Verbal stigma was also described as offensive remarks towards cancer patients like "do not try coming near to me, go away", "you are going to die soon from cancer". The physical or social isolation and verbal abusive comments were also described as a manifestation of stigma in India [13], and in rural Kenya [31].

Unsurprisingly, the stigmatization does have negative impacts on cancer patients on both personal and social levels. Interviewees reported how the marginalization of them from society led to the loss of their social support. A study in Karnataka, India described how the diagnosis of cervical cancer generated a fear inside a cancer patient and their kin whether to disclose their diagnosis to others or not and how they kept the diagnosis within their close family members only. The stigmas related to cervical cancer was a potential barrier to screening and often led to delayed healthcare-seeking behavior among women having symptoms [13]. As described in a study done in America, women felt guilt, embarrassment, and shame as a result of their disease, and it generated a fear inside them about being misjudged by the society upon disclosure of diagnosis, which ultimately led to the loss of their social support [28]. A study in Canada described the discriminative behaviors towards cancer (of any type) survivors in their workplace like discrimination while hiring, harassment, bullying, demotivating them to quit their job, or retire early [32]. Though workplace discrimination is not explored in this study, cervical cancer survivors may face the same consequences at their workplace.

The study adds to the body of research seeking a broader understanding of stigma and its impact on women's personal and social life. As reported by few respondents that with time when the cervical cancer patient got better with treatment, it overturned the thoughts and beliefs of most people towards the stigma related to it. Thus, it shows the importance and needs to implement the awareness programs at an individual and community level to change the way cancer is stereotyped in society.

## Message of survivors

The survivors, though late, realized the importance of regular screening in early detection and treatment of cancer, and changed their perspective towards cancer. Almost all survivors urged other women to seek medical care when they had any concerning symptoms. Cancer is a devastating diagnosis to face and digest and can scare anyone. Forming a group of survivors and providing an opportunity for cancer patients to interact with them can ease resolve the fear of cancer and encourage them to opt for the treatment.

## Study strength and limitation

**Strength.** This is the first study conducted in Nepal to explore the experiences and perspectives of cervical cancer survivors. The qualitative study design itself is the strength of this study. By using this qualitative approach, we were able to assemble the comprehensive understandings of the survivor's perspective of cervical cancer and their journey through this cancer. The conclusion drawn from this study can add a vital element in this research area which can help improve and modify accordingly the awareness campaigns and health care policies, that will implement soon. This study will also help design appropriate interventions and uplift the lifestyle of a cervical cancer survivor, hence accelerate the achievement towards Sustainable Development Goals.

**Limitation.** The study findings cannot be generalized as the chosen sample for the study is purposive and from Chitwan district only and does not reflect the entire female population of Nepal. The knowledge, experiences, and stigma women endured through in the urban settings may be different compared to those experienced by women in the rural setting. We interviewed only those who are alive and volunteered to share their story with us. It would add much more flavor in our study if we add the perspectives of the community and women living with cervical cancer.

## Conclusions

This study demonstrated poor awareness about cervical cancer, delayed healthcare-seeking behavior, and the stigma related to cervical cancer. Thus, there is an urgent need of intervening at multiple levels, at the community level to make women and the public aware of cervical cancer to the federal level for launching effective health education campaigns and making policies for cervical cancer prevention programs. Additionally, the survivors should be included as an integral part of the health policy formation. The implementation of such programs may save the lives of hundreds of women and overturn people's perspective towards cervical cancer thus, saving women from going through all the negative experiences related to cervical cancer. This study also recommends more qualitative studies with the cervical cancer survivors in both urban and rural settings to gain their perspectives, knowledge, experiences, and attitudes about it and add them to our research.

## Acknowledgments

We would like to thank all the cervical cancer survivors and their accompanying persons for their generous contribution to this research. We would also like to thank Dr. Rabin Gautam for helping us revise the manuscript.

## Author Contributions

**Conceptualization:** Gambhir Shrestha, Rashmi Mulmi, Bhola Siwakoti.

**Data curation:** Gambhir Shrestha, Rashmi Mulmi, Bhola Siwakoti.

**Formal analysis:** Gambhir Shrestha, Rashmi Mulmi, Prabin Phuyal, Rahul Kumar Thakur.

**Investigation:** Gambhir Shrestha, Rashmi Mulmi.

**Methodology:** Gambhir Shrestha, Rashmi Mulmi.

**Project administration:** Gambhir Shrestha.

**Resources:** Gambhir Shrestha.

**Software:** Gambhir Shrestha.

**Supervision:** Gambhir Shrestha, Bhola Siwakoti.

**Validation:** Gambhir Shrestha.

**Writing – original draft:** Gambhir Shrestha, Rashmi Mulmi, Prabin Phuyal, Rahul Kumar Thakur.

**Writing – review & editing:** Gambhir Shrestha, Rashmi Mulmi, Prabin Phuyal, Rahul Kumar Thakur, Bhola Siwakoti.

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
