## [Decision Letter · Decision Letter 0]

21 Aug 2020

PONE-D-20-16310

Experiences of cervical cancer survivors in Chitwan, Nepal: a qualitative study

PLOS ONE

Dear Dr. Shrestha,

Thank you for submitting your manuscript to PLOS ONE. After careful consideration, we feel that it has merit but does not fully meet PLOS ONE’s publication criteria as it currently stands. Therefore, we invite you to submit a revised version of the manuscript that addresses the points raised during the review process.

We look forward to receiving your revised manuscript.

Kind regards,

Pranil Man Singh Pradhan

Academic Editor

PLOS ONE

Journal Requirements:

2) Please specify whether an interview guide was used to interview the participant. If yes, please describe and/or include a copy as a Supporting Information file.

3) We suggest you thoroughly copyedit your manuscript for language usage, spelling, and grammar. If you do not know anyone who can help you do this, you may wish to consider employing a professional scientific editing service.  

Reviewers' comments:

Reviewer's Responses to Questions

**Comments to the Author**

1. Is the manuscript technically sound, and do the data support the conclusions?

Reviewer #1: Partly

Reviewer #2: Yes

2. Has the statistical analysis been performed appropriately and rigorously? 

Reviewer #1: No

Reviewer #2: Yes

3. Have the authors made all data underlying the findings in their manuscript fully available?

Reviewer #1: Yes

Reviewer #2: Yes

4. Is the manuscript presented in an intelligible fashion and written in standard English?

Reviewer #1: Yes

Reviewer #2: Yes

5. Review Comments to the Author

Reviewer #1: The topic of research is relevant for Nepal context. This research has potential to contribute to the situation with cervical cancer in the country.

I have some comments which are listed below:

1. Line 48: please correct “low-to-middle-income”

2. Line 59: references are required here.

3. Line 63: references are required

4. Line 64: references are required

5. Line 69: references are required

6. Line 87: How FGD guide was pretested?

Also references which were used for guide development should be provided.

7. Line 93 and 95: Same sentence is repeated twice. Please correct.

8. In the Data collection part of Methods section author should elaborate more on who exactly was conducting FGD, background and previous experience of that person in the qualitative studies.

9. Settings for interview should be mentioned.

10. Line 102: Selection criteria for participants are not clear. What do you mean by “completed the treatment”? It is difficult to decide point for complement of the treatment for cancer patient. Did you select patients with any stages of Ca during diagnosis or some particular stage?

11. As sample size is not calculated for qualitative research, the data saturation usually define number of FGDs which researcher should conduct. This information is missing in the Method part.

12. Data analysis section for qualitative research should be more detailed. Microsoft Office Excel sheets is not used for analysis, should not be mentioned here.

Was analysis done manually or with help of special software for qualitative research analysis?

Author should give an example how exactly themes and codes were extracted form the data. Examples of meaningful units also should be provided. It can be done in the table format.

13. Also authors presented socio-demographic part quantitatively. This shall be mentioned in the data analysis section.

14. Ethical approval part should also mention separately consent for tape recording.

Result part:

15. Table 2: Themes are something broader that it is identified in the table. Fair skin color. Tobacco would be codes here within Theme which can be “Survival reflection …”.

Please check your data analysis process thoroughly. Same for other Themes in the Result section.

16. All tables have double headings. One is written as a table title but it is duplicated in the up row of each table. Please remove.

17. Every quotation of the participants had some numbers in the end after age which is confusing. It is enough to mention age only in the brackets.

18. One of the objectives of this research which is mentioned in the Introduction part was: to measure and intervene to reduce this stigma. This objective is not covered in the result part.

Discussion part:

About half of the Discussion part simply repeating Result section. Please rewrite.

Also Discussion part is too long and confusing. I would suggest making subheadings for discussion.

Thank you

Reviewer #2: Page 4, line 64: "...cured by therapy" - Please say which type of therapy - ? radiotherapy

Page 6, line 90: suggested change - experienced in cervical cancer screening.

Page 7, line 120: "... or thump print". Suggested change - or thumbprint

Page 7, line 125: Suggested change- 53 years (SD 10.8) ranging from 35 to 81 years.

Page 7, line 127: Since all the participants in the study were ever married why not mention that. Suggested change - In our study group all the participants were ever married with currently 88% currently married and 12% widowed.

Page 8, line 138: change have to had

Page 9, line 147: "and" after tobacco

Page 10, line 156: I suggest to put all the symptoms related to vaginal bledding first then other symptoms. One can also make Vaginal bleeding as a main heading and the types as a sub heading.

Page 14, line 234: Suggested change- Cancer as a whole is stereotyped as cause of death and is contagious in the society.

Page 18, line 323:Suggested change- they had prior to diagnosis,

Page 19, line 341: Suggested change- the majority of them had no idea what the cause of cervical cancer in them.

Page 20, line 356: having no symptoms at all and embarrassment that... [remove: the fear of]

Page 24, line 453: Suggested change- though late, realized the importance...

6. PLOS authors have the option to publish the peer review history of their article (what does this mean?). If published, this will include your full peer review and any attached files.

Reviewer #1: No

Reviewer #2: **Yes: **Nishchal Dhakal

---

## [Author Response · Author response to Decision Letter 0]

23 Sep 2020

First of all, we would like to thank the editor and the reviewers for their comments, which have helped significantly to improve the manuscript. We have tried to address all the comments of the reviewers to our possible knowledge. 

Journal Editor:

Response: We have ensured the manuscript meets the PLOS ONE’s requirement.

2) Please specify whether an interview guide was used to interview the participant. If yes, please describe and/or include a copy as a Supporting Information file.

Response: We have described in detail the interview guide in the Methodology section.

3) We suggest you thoroughly copyedit your manuscript for language usage, spelling, and grammar. If you do not know anyone who can help you do this, you may wish to consider employing a professional scientific editing service. 

Response: We have gone through the manuscript rigorously to edit for the language and grammar.

Reviewer #1: 

1. Line 48: please correct “low-to-middle-income”

Response: We corrected it as mainly in low and middle-income countries

2. Line 59: references are required here.

Response: References are added.

3. Line 63: references are required

Response: References are added.

4. Line 64: references are required

Response: References are added.

5. Line 69: references are required

Response: References are added.

6. Line 87: How FGD guide was pretested? Also references which were used for guide development should be provided.

Response: We have not pretested the guide was prepared from already used questionnaires from published articles. The references are added as “An interview guide was prepared by an expert panel of faculty with the help of already published articles [10,13].”

7. Line 93 and 95: Same sentence is repeated twice. Please correct.

Response: Corrected.

8. In the Data collection part of Methods section author should elaborate more on who exactly was conducting FGD, background and previous experience of that person in the qualitative studies.

Response: We have elaborated on the revised manuscript mentioning the person with their responsibility in conducting FGDs along with their background and their experiences.

9. Settings for interview should be mentioned.

Response: We added the settings for the interview.

10. Line 102: Selection criteria for participants are not clear. What do you mean by “completed the treatment”? It is difficult to decide point for complement of the treatment for cancer patient. Did you select patients with any stages of Ca during diagnosis or some particular stage?

Response: By completed, we meant more than two years of treatment for cervical cancer. This has been mentioned in the revised manuscript. The selection of participants has been elaborated as “The inclusion criteria were any cervical cancer survivors residing nearby BPKMCH, who have completed more than two years of cancer treatment. They were selected purposively and contacted via telephone to participate in the study. A total of 17 participants who agreed to participate in the research were invited to BPKMCH for further study.”

11. As sample size is not calculated for qualitative research, the data saturation usually define number of FGDs which researcher should conduct. This information is missing in the Method part.

Response: Only two FGDs were possible as obtaining the cancer survivors within the vicinity of the Cancer Hospital was limited. This is now mentioned in the manuscript.

12. Data analysis section for qualitative research should be more detailed. Microsoft Office Excel sheets is not used for analysis, should not be mentioned here.

Was analysis done manually or with help of special software for qualitative research analysis?

Author should give an example how exactly themes and codes were extracted form the data. Examples of meaningful units also should be provided. It can be done in the table format.

Response: We have rewritten the data analysis part. Excel has been deleted from the analysis part. The analysis was done manually. “After the collection of data, the audio records and notes were translated and transcribed from the Nepali language to the English language by two members of the research team (GS, RM). Transcripts were examined line-by-line and analyzed using the framework approach. The themes were taken based on previous studies. This approach was chosen, as it allows deductive analysis based on our study objectives. Three members of the research team (GS, RM and PP) independently coded the FGD through discussion and agreed on the coding framework. The data was organized into charts. This helped to put the analyses of responses according to the participants. The socio-demographic characteristics of the survivors were presented in frequency, percentage, mean and standard deviation.” The theme summary has been provided in the table 2.

13. Also authors presented socio-demographic part quantitatively. This shall be mentioned in the data analysis section.

Response: The socio-demographic part has been added to the analysis part.

14. Ethical approval part should also mention separately consent for tape recording.

Response: Ethical approval for tape recording is added in line 128.

15. Table 2: Themes are something broader that it is identified in the table. Fair skin color. Tobacco would be codes here within Theme which can be “Survival reflection …”.

Please check your data analysis process thoroughly. Same for other Themes in the Result section.

Response: We have now specifically coded the themes in table 2 and other themes as suggested.

16. All tables have double headings. One is written as a table title but it is duplicated in the up row of each table. Please remove.

Response: We have corrected all the table headings. Duplicates are removed.

17. Every quotation of the participants had some numbers in the end after age which is confusing. It is enough to mention age only in the brackets.

Response: The number was their identification number but it looks not required, hence we removed as suggested.

18. One of the objectives of this research which is mentioned in the Introduction part was: to measure and intervene to reduce this stigma. This objective is not covered in the result part.

Response: We removed the objective “to measure and intervene this stigma”.

19. About half of the Discussion part simply repeating Result section. Please rewrite.

Also Discussion part is too long and confusing. I would suggest making subheadings for discussion.

Response: We have rewritten the discussion part. Subheadings has been added in the discussion.

Reviewer #2: 

Page 4, line 64: "...cured by therapy" - Please say which type of therapy - ? radiotherapy

Response: we corrected it with “Cured by cancer treatment”

Page 6, line 90: suggested change - experienced in cervical cancer screening.

Response: Changed.

Page 7, line 120: "... or thump print". Suggested change - or thumbprint

Response: Changed.

Page 7, line 125: Suggested change- 53 years (SD 10.8) ranging from 35 to 81 years.

Response: Corrected.

Page 7, line 127: Since all the participants in the study were ever married why not mention that. Suggested change - In our study group all the participants were ever married with currently 88% currently married and 12% widowed.

Response: Corrected as per the suggestion

Page 8, line 138: change have to had

Response: Changed.

Page 9, line 147: "and" after tobacco

Response: Changed.

Page 10, line 156: I suggest to put all the symptoms related to vaginal bledding first then other symptoms. One can also make Vaginal bleeding as a main heading and the types as a sub heading.

Response: We arranged vaginal bleeding related symptoms.

Page 14, line 234: Suggested change- Cancer as a whole is stereotyped as cause of death and is contagious in the society.

Response: Changed.

Page 18, line 323:Suggested change- they had prior to diagnosis,

Response: Changed.

Page 19, line 341: Suggested change- the majority of them had no idea what the cause of cervical cancer in them.

Response: Changed.

Page 20, line 356: having no symptoms at all and embarrassment that... [remove: the fear of]

Response: Changed.

Page 24, line 453: Suggested change- though late, realized the importance...

Response: Changed.

Thank you.

---

## [Decision Letter · Decision Letter 1]

22 Oct 2020

Experiences of cervical cancer survivors in Chitwan, Nepal: a qualitative study

PONE-D-20-16310R1

Dear Dr. Shrestha,

We’re pleased to inform you that your manuscript has been judged scientifically suitable for publication and will be formally accepted for publication once it meets all outstanding technical requirements.

Kind regards,

Pranil Man Singh Pradhan

Academic Editor

PLOS ONE

Additional Editor Comments (optional):

Reviewers' comments:

Reviewer's Responses to Questions

**Comments to the Author**

1. If the authors have adequately addressed your comments raised in a previous round of review and you feel that this manuscript is now acceptable for publication, you may indicate that here to bypass the “Comments to the Author” section, enter your conflict of interest statement in the “Confidential to Editor” section, and submit your "Accept" recommendation.

Reviewer #1: All comments have been addressed

Reviewer #2: All comments have been addressed

2. Is the manuscript technically sound, and do the data support the conclusions?

Reviewer #1: Yes

Reviewer #2: Yes

3. Has the statistical analysis been performed appropriately and rigorously? 

Reviewer #1: Yes

Reviewer #2: Yes

4. Have the authors made all data underlying the findings in their manuscript fully available?

Reviewer #1: Yes

Reviewer #2: Yes

5. Is the manuscript presented in an intelligible fashion and written in standard English?

Reviewer #1: Yes

Reviewer #2: Yes

6. Review Comments to the Author

Reviewer #1: (No Response)

Reviewer #2: After the initial comments from the reviewers the suggestion have been incorporated and so the article can be accepted now.

7. PLOS authors have the option to publish the peer review history of their article (what does this mean?). If published, this will include your full peer review and any attached files.

Reviewer #1: **Yes: **Dr. Natalia Oli (MD, MPH, PhD)

Reviewer #2: **Yes: **Nishchal Dhakal

---

## [Editor Report · Acceptance letter]

26 Oct 2020

PONE-D-20-16310R1 

Experiences of cervical cancer survivors in Chitwan, Nepal: a qualitative study 

Dear Dr. Shrestha:

I'm pleased to inform you that your manuscript has been deemed suitable for publication in PLOS ONE. Congratulations! Your manuscript is now with our production department. 

Kind regards, 

on behalf of

Dr. Pranil Man Singh Pradhan 

Academic Editor

PLOS ONE